

# Do mothers also "manipulate" grandparental care?

Mari V. Busch, Sandra Olaisen, Ina Jeanette Bruksås and Ivar Folstad

Department of Marine and Arctic Biology, UiT—The Arctic University of Norway, Tromsø, Norway

Corresponding author
Ivar Folstad, ivar.folstad@uit.no

## ABSTRACT

Paternity uncertainty has proven to be a robust ultimate hypothesis for predicting the higher investment in grandchildren observed among maternal grandparents compared to that of the paternal grandparents. Yet the proximate mechanisms for generating such preferred biases in grandparental investment remain unclear. Here we address two different questions for better understanding the proximate mechanisms leading to the observed bias in grandparental investments: (i) is there a larger emphasis on resemblance descriptions (between grandchildren and grandparent) among daughters than among sons, and (ii) do mothers really believe that their offspring more resemble their parents, that is, the children's grandparents, than fathers do? From questioning grandparents, we find that daughters more often and more intensely than sons express opinions about grandchild–grandparent resemblance. Moreover, daughters also seem to believe that their children more resemble their grandmother than sons do. The latter is, however, not the case for beliefs about children's resemblance to grandfathers. In sum, our results suggest that even in a population of Norwegians, strongly influenced by ideas concerning gender equality, there exist a sexual bias among parents in opinions and descriptions about grandchild–grandparent resemblance. This resemblance bias, which echoes that of mothers biasing resemblance descriptions of newborns to putative fathers, does not seem to represent a conscious manipulation. Yet it could be instrumental for influencing grandparental investments. We believe that a "manipulative mother hypothesis" might parsimoniously account for many of the results relating to biased alloparenting hitherto not entirely explained by "the paternity uncertainty hypothesis."

## INTRODUCTION

As the age of their offspring increases, parents will be favored by natural selection to switch from investing in children to investing in grandchildren (*Hawkes et al., 1998*; *Mace, 2000*; *Croft et al., 2015*). This benefit of kin altruism for indirect fitness may for us (*Homo sapiens sapiens*) have led to an extraordinary long, and for females post-reproductive, life compared to our closely related "cousins" (*Hamilton, 1966*; *Williams, 1957*). Additionally, modern humans have resources easily conserved and also easily transferred between generations. Yet humans exhibit biased grandparental investment.

That is, maternal grandmothers have repeatedly been found to provide more care and resources to grandchildren than paternal grandmothers (*Bishop et al., 2009*; *Busch, 2010*; *Danielsbacka et al., 2011*; *Euler & Weitzel, 1996*; *Fischer, 1983*; *Gibson & Mace, 2005*; *Laham, Gonsalkorale & Von Hippel, 2005*; *Michalski & Shackelford, 2005*; *Pollet, Nelissen & Nettle, 2009*; *Pashos & McBurney, 2008*; see, however, *Pashos, 2000*). Additionally, maternal grandfathers are found to invest more in grandchildren (i.e., only second to maternal grandmothers and on the approximate same level as paternal grandmothers) than paternal grandfathers, which invest the least in grandchildren (*Bishop et al., 2009*; *Euler & Weitzel, 1996*; *Laham, Gonsalkorale & Von Hippel, 2005*; *Pollet, Nelissen & Nettle, 2009*; *Pashos & McBurney, 2008*). Although some of the observations may be confounded (*Tran, Fisher & Voracek, 2009*), they show, with few exceptions (*Pashos, 2000*), a clear bias toward a larger grandparental investment among maternal grandparents than among paternal grandparents in a diverse set of cultures.

Derived from evolutionary theory, the "paternity uncertainty hypothesis" (*Smith, 1988*) is the main ultimate explanation for biased grandparental investment in grandchildren. Indeed, male mate guarding, rather than paternal care, seems to have been the driver of the evolution of monogamy in humans, as it secures a partner and ensures paternity certainty in the face of more promiscuous competitors (*Schacht & Bell, 2016*). The "paternity uncertainty hypothesis" relates to the uncertainty of paternity (which may be especially pronounced in species with concealed ovulation and internal fertilization) the investing grandparent has to the genetic relatedness of grandchildren. Maternal grandmothers can be sure of their genetic relationship to their children and grandchildren, but maternal grandfathers and paternal grandparents cannot under natural circumstances be entirely certain of their genetic relationship to children and grandchildren because of non-paternity (i.e., when the putative father is not biological father). There are no uncertain intergenerational genetic links between a child and its maternal grandmother, one uncertain genetic link between a child and its maternal grandfather and its paternal grandmother (through the child's putative father), and two uncertain links between a grandchild and the paternal grandfather. Based on this, the maternal grandmother is predicted to invest the most in grandchildren and the paternal grandfather is predicted to invest the least. Maternal grandfathers and paternal grandmothers are equally uncertain and are expected to invest more than paternal grandfathers, but less than maternal grandmothers. The paternity uncertainty hypothesis has proven robust in predicting the observed asymmetric grandparental investments (*Bishop et al., 2009*; *Danielsbacka et al., 2011*; *Euler & Weitzel, 1996*; *Laham, Gonsalkorale & Von Hippel, 2005*; *Pollet, Nelissen & Nettle, 2009*; *Pashos & McBurney, 2008*; *Danielsbacka & Tanskanen, 2012*; *Pashos, 2000*), but the proximate mechanisms leading to such bias are still largely unknown.

Most likely caused by the advent of modern contraceptive methods and the acceptance of hygienic selective abortions, the non-paternity rates have declined from approximately 8–10% in the 1930s to about 1–3% in modern industrialized western populations (*Anderson, 2006*; *Larmuseau et al., 2013*; *Voracek, Haubner & Fisher, 2008*; *Wolf et al., 2012*;

*Bellis et al., 2005*; *Larmuseau et al., 2017*; *Larmuseau, Matthijs & Wenseleers, 2016*). Yet as our mental concepts of non-paternity due to infidelity are not updated to the current reproductive environment, subjective rates of non-paternity among males in the same modern populations are as high as 10% (*Voracek, Fisher & Shackelford, 2009*; *Russell & Wells, 1987*); a number corresponding to that of non-paternity observed within present hunter-gatherer societies (*Neel & Weiss, 1975*). Thus, old mental adaptions from our past evolutionary reproductive environment, that is, the environment of evolutionary adaptiveness, still evoke relatively high sensitivity to possible cuckoldry and consequently non-paternity among putative fathers and may similarly also influence grandparental motives. Interestingly, women in western societies, who presumably should have better knowledge about female infidelity than men, provide non-paternity estimates as high as 14% (*Voracek, Fisher & Shackelford, 2009*).

Since grandparental investment is a limited resource, grandparents that are able to separate kin from non-kin would be supporting their own rather than foreign genes in future generations, and consequently be favored by natural selection. One might expect that physical similarity is the most useful cue for kin recognition. This has repeatedly been documented in father–offspring relationships (*Alvergne, Faurie & Raymond, 2007*; *Alvergne et al., 2009*; *Bressan & Dal Martello, 2002*; *Daly & Wilson, 1982*; *Kaminski et al., 2009*) and a positive relationship between children's resemblance to the father and the father's investment in children have also been documented (*Alvergne, Faurie & Raymond, 2007*; *Chang et al., 2010*; *Alvergne, Faurie & Raymond, 2009*; *Apicella & Marlowe, 2004*; *Platek et al., 2002*). Conversely, paternal neglect and physical abuse have been associated with reduced resemblance between father and child (*Bellis et al., 2005*; *Burch & Gallup, 2000*; *Alexandre et al., 2011*).

As a person's own subjective experience of resemblance (which originally may have been established from evaluating kin) may be influenced by self-deceptive (adaptive) wishful thinking (*Trivers, 2011*; *Bressan & Dal Pos, 2012*), one should also pay attention to the opinion of others. Such opinions are often readily available, as children's resemblance to family members is often commented on (*Daly & Wilson, 1982*; *McLain et al., 2000*). Yet in our verbally sophisticated species, ascriptions of resemblance by others are not entirely reliable, and mothers and their relatives seem to manipulate such descriptions (*Daly & Wilson, 1982*; *McLain et al., 2000*; *Regalski & Gaulin, 1993*). That is, mothers are more likely to ascribe resemblance of newborns to the domestic father than to themselves, and this bias is exaggerated in the presence of fathers and reduced in their absence (*McLain et al., 2000*). Moreover, objective judges match photographs of focal mothers more often to their newborns than they match photographs of putative fathers to the newborns. Consequently, the bias in how mothers remark resemblance clearly does not reflect actual resemblance and may be an evolved or conditioned response to assure domestic fathers of their paternity (*McLain et al., 2000*). It is not unlikely that mothers in a similar way may also communicate deceptive judgments to grandparents about grandchild–grandparent resemblance and thus influence grandparental investments in grandchildren. It should be noted that such communication need not involve conscious deception—it need not be premeditated.

Paternity uncertainty has recently been included in a larger model emphasizing parent's and alloparent's benefits from investing in the mother as well as in her children (*Perry & Daly, 2017*). That is, as mothers most often are prime parental caregivers, parenting targeted toward mothers rather than toward fathers gives the highest payoff in residual nepotistic value and, in turn, fitness. The explanation rests on an asymmetry in fitness payoffs for parents and alloparents influencing their decision making during resource allocations to mothers, fathers and their children. Although the two models (i.e., paternity uncertainty and the sexual bias in residual nepotistic gains) emphasize different routes to matrilateral bias in parental investments, the ultimate models are not mutually exclusive and may both be at work simultaneously.

In the two within-family studies reported here we focus on a more proximate alternative. That is, we examine how daughters and sons may differ in verbal communication of their children's resemblance to grandparents. Is it possible that mothers more often and more intensely than fathers communicate information about grandchildren's familiar resemblance, not unlike that seen when mothers ascribe newborns resemblance to fathers? Further, if this is the case, is this difference in communication of information rooted in sexual differences in beliefs about resemblance? In our first study, we asked grandparents if their son and daughter had behaved differently when ascribing grandchildren's resemblance to them. In our second study, we examined whether mothers and fathers differed with regard to beliefs about grandchild–grandparent resemblance. The two studies were conducted in a population exposed to long-term conscience rising about gender equality and should be considered conservative.

# STUDY 1
## Materials and methods
### *Questionnaire*

We constructed a questionnaire (see Supplementary Material) to examine how grandparents perceived their grandchildren's resemblance to them, and if they had experienced comments on grandchildren's resemblance to them from their sons or daughters. The questionnaire was either read as an interview between one of us (IJB or SO) and the grandparents, or sent to the grandparents who filled in their answers. For simplicity, the questioning focused on the grandparent's son's and daughter's firstborn child. If grandparents had more than one son or more than one daughter with own children, they were asked to focus on their first son and their first daughter that got children when answering the questions. They were initially asked how similar they felt the two grandchildren were to them, with separate seven-point scales ranging from 1 (no resemblance) to 7 (very clear resemblance) for both physical and psychological resemblance. Physical resemblance was defined to the grandparents as physical characteristics like facial features, body shape, posture etc., and the psychological resemblance was defined as similarity in behavior, personality, talents and thought patterns. Time is a finite resource that can be invested unequally among different items.

One way of assessing asymmetric grandparental investment is to use the reported frequency of contact between grandparents and their grandchildren as a measure of

investment (*Pollet, Nelissen & Nettle, 2009*). Contact between grandparent and grandchildren was measured through questions on frequencies of face-to-face contact, as a cue for their true investment of time. The grandparents were also asked whom they felt most often initiated grandchildren–grandparent contact, and to give estimates of traveling distances and expenses. In addition, grandparents were asked to report whether their daughter and son was living with their grandchild's father or mother, respectively. Toward the end of the interview/questionnaire, grandparents were asked whether their daughter, their son, or other family members had ever made comments about the grandchildren's resemblance to the grandparents; the nature of these comments (positive or negative regarding resemblance) and how intensely these were made. All grandparents were Caucasian. It should also be noted that Norway has a governmentally supported health care system, an average pension age of 67 years and a well-developed governmental child-care, including a financial support system.

### Participants

To be included in the study, respondents had to be both a maternal and paternal grandparent (i.e., they had to have at least two biological grandchildren, one through a son and one through a daughter). The grandparents consented to participate after being informed that their answers were to be used in a paper on resemblance and relations between grandparents and grandchildren. All data collection was anonymized and ethics approval was consequently not needed. A total of 87 grandparents, 64 grandmothers and 23 grandfathers, with no relation to each other, answered the questionnaire (one did not report on resemblance descriptions). Most of the respondents were sampled haphazardly from various public locations in Northern Norway, such as the University Hospital of North Norway and shopping malls in the towns Tromsø and Narvik. Some respondents were also recruited through family and friends in Narvik, Lofoten, Mosjøen, Sortland and Steigen. The mean age of grandparents was 65 years (range: 54–89; eight grandparents did not report their age). The average age was 45 years for both their first daughters to have children (range: 25–62), and for their first sons to have children (range: 28–64). For the firstborn child of the respondents' daughters, the mean age was 20 years (range: 6 months–44 years), where 37 of them were boys, and 50 girls. For the son's firstborn child, the mean age was 17 (range: 6 months–40 years), of which 43 of them were boys, and 46 were girls. Although our hypothesis gives clear predictions about the directions of the outcome for the male–female comparisons, *p*-values are reported two-tailed throughout. Yet we have not adjusted for multiple comparisons (*Nakagawa, 2004*; *Ioannidis, 2018*).

### Results

There was no sex difference in the age of parents ($F_{172} = 0.28$, $p = 0.82$) or in the age of their firstborn child ($F_{168} = 0.46$, $p = 0.10$). Moreover, the traveling distance (given as duration of travel) to the firstborn of daughters was reported by 44% of the respondents to be less than 30 min, between 30 min and 4 h for 23%, and more than 4 h for the remaining 33%. For the firstborn of sons, the traveling distance was less than 30 min for 44%,

30 min to 4 h for 11%, and more than 4 h for 45% of the grandparents. There was no significant difference in traveling distance to daughter's or son's firstborn ($\chi^2 = 4.80$, $p = 0.09$). A total of 66% of the grandparents estimated a cost of 500 Norwegian Kroner (NOK) or less in traveling expenses for reaching their daughters' firstborn, whereas 9% estimated a cost between 500 and 1,500 NOK, and 25% estimated more than 1,500 NOK. For their sons' firstborn, 56% estimated a cost less than 500 NOK, 15% between 500 and 1,500 NOK, and 29% more than 1,500 NOK. No significant difference was found in traveling expenses to son's or daughter's firstborn child ($\chi^2 = 1.98$, $p = 0.37$).

More grandparents reported having more frequent face-to-face contact with their daughter's child than their son's child (daily or weekly: 45% vs. 29%; monthly to every sixth month: 46% vs. 61%; once a year or less: 9% vs. 10%, respectively). That is, in the role as maternal grandparents, grandparents have slightly more frequent contact with their grandchildren than in the role as paternal grandparents. Although this does indicate behavioral differences between maternal and paternal grandparents, no strict statistical differences in contact frequencies could be documented ($\chi^2 = 4.94$, $p = 0.08$). For visits to their daughter's child, 22% of the grandparents said they were most often the initiator of visits, whereas 39% said their grandchild initiated visits. Moreover, 30% of the grandparents said the visits were initiated equally, and the remaining 4% were not sure. For visits to their son's child, 25% of the grandparents said they most frequently initiated the visits, 38% answered their grandchild did, 33% said both and 3% were not sure. That is, differences in grandparental initiations of visits to the daughters' and the sons' child was not significant ($\chi^2 = 0.39$, $p = 0.94$). Additionally, more daughters than sons were reported by the grandparents to live together with their child's parent (75% vs. 63%), but the difference was not significant ($\chi^2 = 2.03$, $p = 0.15$).

Grandparents reported their daughter's child to resemble them slightly more than their son's child for both physical and psychological resemblance. That is, mean rating for physical resemblance between grandparent and the child of a daughter was 3.4 (SD = 1.7), while mean rating for the son's child's resemblance to the grandparent was 3.0 (SD = 1.7). This difference was not significant ($\chi^2 = 5.45$, $p = 0.49$). The respective values for psychological resemblance was 4.0 (SD = 1.6, for the daughter's child), 3.5 (SD = 1.6, for the son's child) and $\chi^2 = 5.61$, $p = 0.47$. However, when grandparents were asked in a force choice manner whether their son's or their daughter's child resembled them the most, the results were in favor of the child of their daughter, both for physical ($\chi^2 = 10.6$, $p = 0.001$, $n = 48$ daughters' child, $n = 21$ sons' child) and psychological resemblance ($\chi^2 = 5.71$, $p = 0.02$, $n = 45$ daughters' child, $n = 25$ sons' child).

When asked if any family members had mentioned anything about their grandchildren's resemblance to them, 59% ($n = 49$) of the grandparents confirmed this. Sisters, brothers, uncles, aunts, great grandmothers, and in-laws were the ones who had made comments on grandchild–grandparent resemblance. 49% ($n = 24$) of the comments had been made about the daughter's child, and 51% ($n = 25$) about

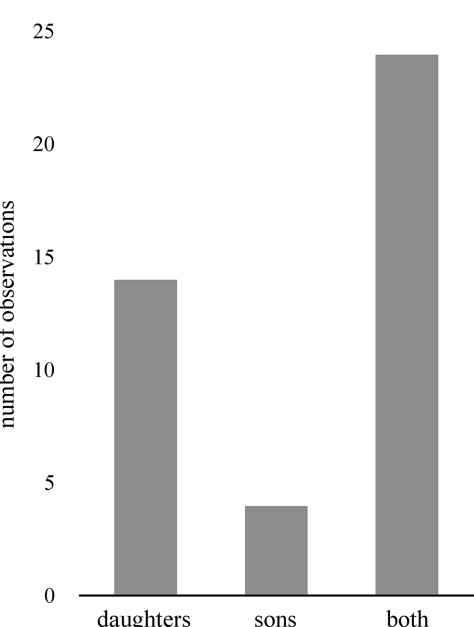

Comments about resemblance

**Figure 1 Comments about resemblance.** Bar plot showing the number of grandparents reporting if their daughter, their son, or both had ever made comments to them about their grandchildren resembling them.               

the son's child. However, when asked whether they could recall their son or daughter ever emphasizing their grandchildren's resemblance to them, the results were quite different. Of the 42% of the grandparents who remembered that their children had talked about resemblance, 33% of them reported that only their daughter had mentioned the topic, while 10% of them reported their son to be the only one to have made comments on resemblance (exact binomial test, $p = 0.03$, $n = 4$ sons, $n = 14$ daughters, Fig. 1). When only including grandfather's recollection about who made comments, the difference was still apparent ($\chi^2 = 5$, $p = 0.03$, $n = 5$ daughters and zero sons).

The remaining 57% of the grandparents, who remembered their children emphasizing resemblance, reported that both their son and daughter had made such comments. These grandparents were asked whom they felt most frequently or intensely made such comments. A total of 54% said their daughter, 38% said they could not recall any difference between sons and daughters, and only 8% said their son more often and more intensely expressed opinions about grandchild–grandparent resemblance (exact binomial test, $p = 0.007$, $n = 2$ sons, $n = 13$ daughters, Fig. 2). Thus, daughters significantly more often than sons, intensely reminded their parents about grandchildren's resemblance to them.

Grandparents were also asked whether their son or daughter had ever stated that their siblings' child did not resemble them, and as many of 93% of them could not recall any of them mentioning this. Of the remaining grandparents, 3% had experienced only their daughter making such comments, 1% only their son and 2% had heard comments from both of them. No meaningful difference was found for these results (exact binomial test, $p = 0.63$, $n = 1$ son, $n = 3$ daughters).

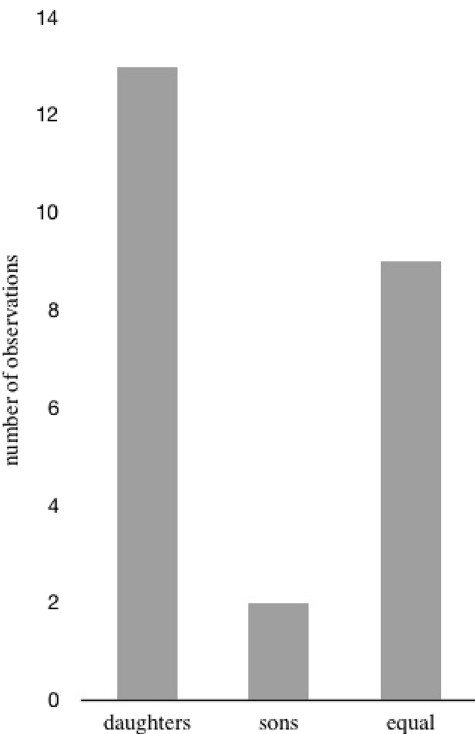

Figure 2 **Strongest opinion on resemblance.** Bar plot showing whom the grandparents felt most frequently or intensely argued resemblance between grandparent and grandchildren when reporting that both son and daughter had talked about resemblance.   

# STUDY 2

## Materials and methods

### Questionnaire

We constructed a questionnaire (see Supplementary Material) to examine parents' perceived resemblance to and between close kin. In this study, close kin are children, parents, siblings (restricted to the first four), nieces and nephews. All parents were asked to categorize the term resemblance as resemblance in personality (i.e., psychological) or in appearance (i.e., physical) before answering the questions concerning resemblance to and between kin. This is because the term resemblance does not necessarily make a distinction between the two, and psychological and physical resemblance might weigh differently for investment decisions made by males and females. That is, fathers may be more influenced by physical resemblance and less influenced by psychological resemblance than mothers (*Heijkoop, Dubas & Van Aken, 2009*). Parents were further asked to give the age and sex of their child/children and sibling/siblings. In addition, they were told to estimate, on a seven-point scale ranging from 1 (< yearly) to 7 (daily), their child/children's current frequency of contact with their mother and father (i.e., respondent's parents), respectively. In the second half of the interview, questions were asked in a forced choice manner. Parents were asked to answer which one's children they thought resembled the respondent's own mother and father the most, their

own children or their sister's/brother's children. In a similar manner, parents were asked whether they saw their child/children as resembling them more or less then their sibling's children resembling their sibling. Respondents having more than one sibling of the same and/or opposite sex, which also had children, were instructed to make the comparisons to the sibling closest in age.

*Participants*

Males and females of the scientific staff, all with Scandinavian sounding first and surnames, at various faculties at the UiT—The Arctic University of Norway, were opportunistically contacted on their office telephones during working hours by one of the authors (MVB). The potential respondents were informed that the authors were writing a paper on family resemblance, after which they were asked to participate in a telephone interview on the subject. In order to make the sampling as random as possible, no information about the staff, other than name (i.e., sex), office telephone number and position at the University, was known prior to contact. A total of 51 consenting participants (21 males and 30 females; mean age ± SD = 50.26 ± 10.85, 50.23 ± 7.98, respectively) met the criteria of having at least one biological child and one sibling of the opposite sex also with at least one biological child. The average age of firstborns among female parents was 20.7 years (SD = 9.5) and 17.7 years (SD = 2.7) among male parents. All data were anonymized and, according to the Norwegian Data Protection Office for Research (NSD), no ethical approval was needed. Male participants had a mean ± SD of sons = 0.91 ± 0.89, and daughters = 1.14 ± 0.62. Female participants had 1.1 ± 0.96 sons, and 1.1 ± 0.85 daughters. The respondents were asked to estimate the current frequency of contact between their children and the respondent's parents. This required that the grandparents in question were still alive, which was not the case for all respondents. Due to this and because not all participants had a sibling of the same sex or were able to answer every question, the sample size varies for the different analyses and will be given for each. Again, $p$-values are reported two-tailed.

## Results study 2

There was no sex difference in the age of the respondent's children ($F_{49}$ = 3.3, $p$ = 0.33). Moreover, mothers and fathers in this sample did not differ in categorizing resemblance to resemblance in personality, appearance, or both ($\chi^2$ = 0.02, d$f$ = 2, $p$ = 0.99). A total of 13 (61.9%) males and 19 (63.3%) females reported that they found resemblance in both personality and appearance to be equally important when assessing resemblance. Only two (9.5%) males and three (10.0%) females stated that resemblance in appearance was most important, and six (28.6%) males and eight (26.7%) females found resemblance in personality as most important in assessment of resemblance.

Females had a tendency to report their children's contact with the children's maternal grandmother as more frequent than what was reported by males regarding their children's contact with paternal grandmother ($t$ = −1.35, d$f$ = 32, $p$ = 0.19, $n$ = 15 males and $n$ = 19 females). The frequency of contact between children and the respondent's father

was not differently reported by males and females ($t$ = 0.74, d$f$ = 19, $p$ = 0.47, $n$ = 7 males and $n$ = 14 females).

When asked whose children resemble the respondent's mother the most, the respondent's children or the children of an opposite sexed sibling, females reported greater resemblance between their own children and their mother (the children's maternal grandmother) compared to what males did (reporting resemblance to the children's paternal grandmother) (Fig. 3). The reversed pattern, though not significant, was found when the respondents were asked to assess their children's resemblance to grandmother relative to the children of a sibling of the same sex ($\chi^2$ = 1.6, $p$ = 0.21, $n$ = 24). That is, both males and females reported own children to bear more resemblance to the children's grandmother (the respondent's mother) compared to a brother's children, but not when compared to a sister's children.

When reporting resemblance between children and the children's grandfather (the respondent's father), there was no difference between males and females, neither when comparing own children to the children of a sibling of the opposite sex ($\chi^2$ = 0.11, $p$ = 0.74, $n$ = 46), nor when comparing own children to the children of a sibling of the same sex ($\chi^2$ = 0.33, $p$ = 0.57, $n$ = 19). Both males and females, though, stated that their children were more similar to their father (the grandchildren's paternal and maternal grandfather, respectively) than a sibling's children, irrespective of the siblings' sex. For resemblance to the children's grandfather, 17 out of 26 ($\chi^2$ = 2.77, $p$ = 0.10) females stated that their children resembled their grandfather more than their brother's children did, and 14 out of 20 ($\chi^2$ = 2.77, $p$ = 0.10) males reported the same for their children's resemblance to grandfather when compared to a sister's children. These results are comparable to the answers given by the respondents when asked which one of themselves and their sibling share most resemblance with their children. Most males and females stated that there was more resemblance between themselves and their own child (children) than between an opposite sexed sibling and his/her child (children) (for females: 20 vs. 4 in favor of own resemblance; for males: 12 vs. 7 in favor of own resemblance).

## DISCUSSION

### Our materials and methods

Females are overrepresented in our samples of both parents and grandparents. Yet as mothers and fathers do not differ in categorizing the term resemblance, answers given to the questions concerning resemblance in the present study seem to reflect similar modes of underlying evaluations for male and female respondents. Moreover, while it might be argued that such resemblance assessment between generations could be confusing for the respondents, the questions in the present two questionnairs were constructed in such a way that the respondents made only a single comparison for each question. It is therefore likely that the recorded answers reveal the respondents' opinions. Yet whether and to what extent these opinions are constrained by cultural correctness or evolved mental constructs is difficult to assess. The many observations of no sexual difference throughout both studies, followed by a sudden appearance

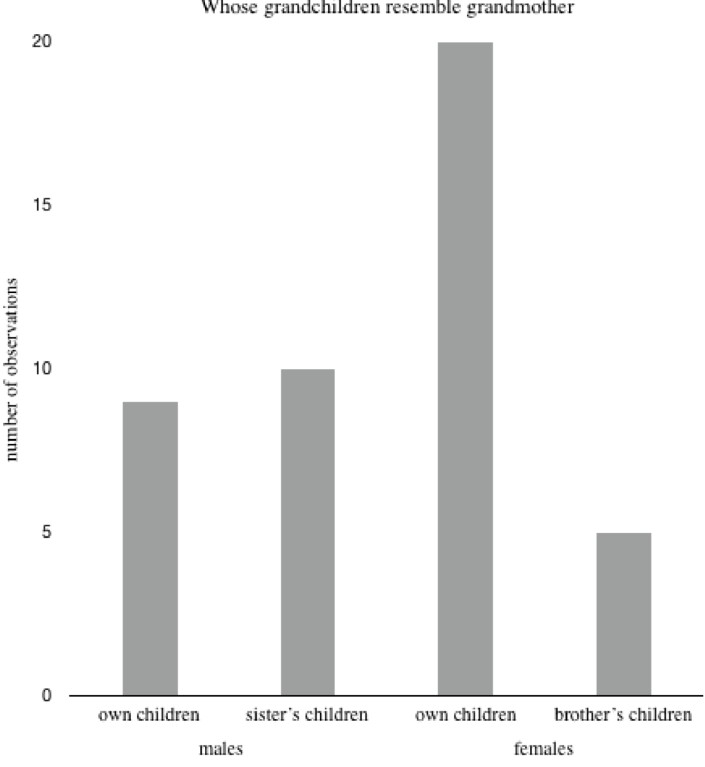

**Figure 3 Whose grandchildren resemble grandmother.** The number of males and females reporting whose children resemble grandmother (respondent's mother) most; own children or the children of a sibling of the opposite sex (i.e., males comparing their children to a sister's children, and females comparing their children to a brother's children, Chi-square = 5.11, $p$ = 0.024, $n$ = 19 males and $n$ = 25 females).

of sexual difference only under forced choice questioning might, however, hint to an influence of cultural correctness over self-reported opinions. Moreover, the reluctance to participating and answering some of the questions (own observations) adds to the latter.

## Our results

Although both sons and daughters potentially could benefit from emphasizing resemblance between grandparents and own offspring (*Schlee & Kirchengast, 2015*), daughters more than sons seem to voice opinions about grandchild–grandparent resemblance in our sample. Daughters also voice opinions more intensely than sons do. Additionally, mothers, more than fathers, seem to believe that their children resemble their mother—the child's maternal grandmother and prime grandparental investor (*Euler & Weitzel, 1996*; *Pashos & McBurney, 2008*). The latter sex-difference in resemblance opinions is, however, not evident for descriptions of grandchild–grandparent resemblance.

Although belief in genetic relatedness has been found to affect resemblance ratings in a positive direction (*Bressan & Dal Martello, 2002*; *Oda, Matsumoto-Oda & Kurashima, 2005*), it could be argued that an increased contact rate and intimacy between maternal

grandparents and grandchildren could lead daughters to report a larger resemblance between their children and their familiar grandparents than sons do. Under this scenario, increased contact leads to a sense of increased resemblance due to familiarity rather than vice versa. Yet although our present questioning of parents and grandparents suggest that grandparents may have a slightly higher frequency of contact with daughters' children than with sons' children, and that grandparents may also show small differences in initiating contact with daughter's rather than son's children, none of these differences reach statistical significance. Additionally, grandparents do not live closer to their daughter's than their son's firstborn, and there seem not to be differences in expenses for travels to sons or daughters. In our sample of grandparents, there is also no difference between son's and daughter's firstborn in whether they still live with the grandchild's biological parent. That is, given equal opportunity for cohabitation with biological children, sons and daughters would experience similar time-periods when children live with a biological parent that is not the child of the focal grandparent. Moreover, there is no systematic age-bias in our sample, suggesting a longer exposure time to grandchildren of daughters rather than sons. In sum, this suggests that opportunities for grandparental contact with grandchildren in our sample are not more constrained for children of sons than for children of daughters. Additionally, when questioning parents about contact rates between their children and their grandparents, we find no significant sex differences. Although many of the comparisons listed above show a slightly larger potential for contact between maternal than paternal grandparents, none of them alone reach effect sizes comparable to those found for sexual differences in parental resemblance descriptions. That is, compared to the measured differences in indices of intimacy reported in our samples (i.e., grandparent reported contact rate, their initiations of contact, settlement distance between grandparent and grandchildren, cohabitation of parents and parental reported grandparent–grandchild contact), there are large sex-differences in grandparental descriptions and opinions on resemblance. This suggests that the reported resemblance bias may partly be established independent of differences in contact rates. Still, it cannot be excluded that the effects of a slightly larger potential for contact between maternal grandparents and grandchildren could be causing the sexual difference in parental descriptions and opinions reported by grandparents. In that case, one would suspect that resemblance descriptions were reflected in the response from grandparents. The unconstrained response from grandparents does not suggest that grandparents felt any resemblance bias toward daughter's children over son's children, that is, grandparents did not report that daughter's firstborn resembled them psychologically or physically more than son's firstborn. Still, when constrained, and force-choice questioned about whether grandparents felt their son's or their daughter's child resembled them the most, the results were definitely in favor of the child of their daughter. We believe that the constriction under our questioning lead the respondents to deviate from their default politically "correct" response, to reporting their "real" opinion. In sum, the presence of our documented sex-specific bias in resemblance-beliefs is interesting independent of the causal pathway for its establishment (i.e., whether it is nurture or nature), as it

might both cause and reinforce asymmetric investment decisions by all four grandparents.

Grandparents reported that daughters more often than sons talked about grandchildren's resemblance. Additionally, when both their daughter and son had made comments about the subject, the daughter was described as the most intens, making comments more frequently than her brother. Most of the participants in this survey were grandmothers (74%), so these findings are not surprising considering that the subject might be considered "girl-talk material." Daughters could therefore be expected to talk more about resemblance to their mothers than sons do. However, using only the grandfather's reported answers, the difference was still significant in "favor" of the daughters. Thus, these results are better explained by the potential influence of differences in beliefs held and reported by mothers and fathers. Our present results also correspond closely with results from a study of 177 female and 56 male Norwegians, having siblings of the opposite sex with children. In this study, where subjects where recruited among parents with kids in kinder-gardens, both males and females reported that sisters more often and more intensely commented on resemblance of own and others children than brothers (*Fisktjønmo, 2017*). Clearly, resemblance descriptions are not "girl-talk material" only. Our finding of a larger emphasis on resemblance descriptions among daughters than sons observed by both grandfathers and grandmothers correspond with the documented sexual differences in resemblance beliefs, and it also clearly echoes results from studies documenting that mothers are more likely to emphasize their newborn's resemblance to the putative father when he is present, than when absent (*Alvergne, Faurie & Raymond, 2007*; *McLain et al., 2000*). Although the latter results only apply to putative fathers, it cannot be excluded that the underlying mental modules may also be operating in relation to grandparental resemblance descriptions.

## The larger perspective

Several studies have indicated that males, due to paternity uncertainty, should rely more heavily on resemblance to children, as a cue of genetic relatedness, for investment decisions than females (*Daly & Wilson, 1982*; *Alvergne, Faurie & Raymond, 2009*; *Apicella & Marlowe, 2004*; *Platek et al., 2002*; *Burch & Gallup, 2000*; *McLain et al., 2000*; *Heijkoop, Dubas & Van Aken, 2009*), but few have extended and investigated this prediction regarding grandparental investment (see, however, *Euler & Weitzel, 1996*; *Schlee & Kirchengast, 2015*; *Pashos & McBurney, 2008*). Also, putative relatedness weighted by certainty in relatedness, better predict willingness to invest in kin than putative relatedness alone (*Antfolk et al., 2017*), and kin selection theory is also supported by studies comparing investments by grandparents and step-grandparents (*Pashos, Schwarz & Bjorklund, 2016*; *Gray & Brogdon, 2017*). In fitness terms, grandparental investments can be viewed as an extension of parental investment, or as *Euler & Weitzel (1996)* points out, as a "differentiated subset of parental effort." Consequently, investment decisions made by grandparents should be subjected to the same scrutiny as paternal investment decisions. Assuming that grandparents are sensitive both
to the belief in genetic relatedness and influenced by what others tell them regarding resemblance to kin, it is not unlikely that their children can influence their investment decisions. Females, who are known to bias resemblance descriptions in their favor (*Alvergne, Faurie & Raymond, 2007*; *McLain et al., 2000*), may indirectly raise doubt about a brother's paternity by emphasizing both their own resemblance and their mother's resemblance to their children. This could, in turn, affect their mother's investment decisions and explain why maternal grandmothers repeatedly appear as the prime grandparental caregivers. It is, however, important to emphasize that this sexual difference in belief and vocal behavior does not have to involve conscious manipulation. It need not be premediated, but rather an evolved unconscious response. Additionally, mothers could also influence their children and it is, in this perspective, interesting that grandchildren are found to rate their resemblance to maternal grandmothers as higher than to maternal grandfathers (*Euler & Weitzel, 1996*). The alternative approach to resemblance descriptions, that is, to increase dissimilarity descriptions of a sibling's children, seem to be less prevalent in our present sample (see also *Fisktjønmo, 2017*), maybe because dissimilarity descriptions may be considered un-polite, and consequently carry costs to the signaler. Moreover, reporting such behavior might also be less acceptable—especially among Norwegians.

Although there might certainly exist cultural effects on decision making among all parents, our "manipulative mother hypothesis" (i.e., that mothers mentally exploit the alloparenting environment by expressing biased resemblance descriptions) could explain many of the results not entirely predicted by the "paternity uncertainty hypothesis" alone (see *Pashos, 2017*). For example, the higher grandparental investments observed among maternal grandfathers compared to that of the equally uncertain paternal grandmothers (*Euler & Weitzel, 1996*), the larger caregiving toward nieces and nephews observed among maternal aunts compared to that of paternal aunts (*Pashos & McBurney, 2008*) and the maintenance of a high matrilateral caregiving among orthodox Jews under high paternity certainty (*McBurney et al., 2002*) are all results that correspond to a larger manipulation of the alloparenting environment among mothers than fathers. Even among rural Greeks, where a patrilateral bias in caregiving is present with the paternal grandmother as the largest investor (*Pashos, 2000*), one cannot exclude that maternal manipulation can be at work. Yet all except the latter of these results could also be explained if alloparental investments for increased residual nepotistic value of mothers gives larger fitness benefits to grandparents than similar investments in fathers (*Perry & Daly, 2017*). Our more parsimonious model, which does not rest upon the above assumption, could account for all the above findings. Additionally, the "manipulative mother hypothesis" might throw new light on why parenthood improves emotional closeness and contact rate between daughter and mother (*Tanskanen, 2017*; *Danielsbacka, Tanskanen & Rotkirch, 2017*), on why emotional closeness and intimacy between sons and mothers decrease after the birth of their first child (*Tanskanen, 2017*) and, not least, on why the relationship between daughter-in-law and mother-in-law has repeatedly been reported to be the least healthy of all parent–grandparent relationships (*Sherlip & Stricker, 1998*; *Euler, Hoier & Rohde, 2009*;

*Fingermann, 2004*; *Lee, Spitze & Logan, 2003*) and thus of potential importance for grandparental support (*Michalski & Shackelford, 2005*). Our hypothesis rests firmly on the repeated findings of deceitful verbal assurance given by mothers to putative fathers (*Alvergne, Faurie & Raymond, 2007*; *Daly & Wilson, 1982*; *McLain et al., 2000*), and future Functional Magnetic Resonance Imaging evaluations could be able to examine whether the same cognitive process domains are involved in both "manipulative" actions. In a larger picture, it also conforms to the slight tendency toward a larger use of indirect aggression, for example, gossiping, among females compared to males (*Archer & Coyne, 2005*; *Card et al., 2008*) and to the repeatedly observed, and sometimes quite large negative effects from reproductive competition among human females (*Lahdenperä et al., 2012*; *Cant & Johnstone, 2008*; *Hammel & Gullickson, 2004*; *Pettay et al., 2016*)—even within kin groups (*Mace & Alvergne, 2012*; *Sear, 2008*; *Willführ, Johow & Voland, 2018*; *Ji et al., 2013*). In sum, the "manipulative mother hypothesis" alone, or in some combination with the "paternal uncertainty hypothesis" and the *Perry & Daly (2017)* model, might account for many of the results that hitherto have been hard to explain parsimoniously. The hypothesis gives clear predictions (e.g., when mothers have restricted access to the alloparenting environment the biased alloparenting should be reduced), and it could consequently easily be falsified.

## CONCLUSION

This study is limited to only include a sample of academics and grandparents from various locations in North-Norway. Yet as Norwegians in general, and in particular Norwegian academics, have been exposed to conscience rising about gender equality, these results should be considered conservative. It is thus plausible that the grandparent's higher perceived resemblance to their daughter's children, daughter's stronger beliefs about children's resemblance to their mother, and the daughters larger extent of resemblance descriptions is not unique for grandparents and parents from Norway. Additionally, the findings may provide a mechanistic, proximate explanation for asymmetric grandparental investment where paternal grandparents are usually found to invest less in grandchildren than maternal grandparents—especially the maternal grandmother. Our study, as any one study, is in no way exhaustive, and whether our findings will stand the scrutiny of future meta-analysis on lager data, remains to be evaluated. Yet we hope our ideas might inspire researchers to explore previously unimagined avenues.

## ACKNOWLEDGEMENTS

Our model is controversial and has been difficult to publish. Therefore, a great thanks to the editor for allowing us to present our contribution and to the two reviewers (the anonymous and Denson McLain) for constructive critics and excellent ideas. Thanks also to Ivar's sister and mother of own kids who initiated these investigations by stating, in the presence of their mother: "Ivar, your kids don't look like you." Thanks also to Joakim Carlsen, Hella Veierud Busch, Marit Folstad, Bill Olaisen and Helen Thoresen for encouragements, to Alexander Pashos for his advice in the planning phase of parts of the study, to Harald A. Euler for reading through an earlier draft

of the manuscript and for his constructive feedback and to Ole Petter Laksforsmo Vinstad for discussions and linguistic improvements at a late stage. Although it is impossible for us to thank those of the scientific staff at various faculties at the University of Tromsø and grandparents in different parts of Northern Norway for participating in the survey, we hope our small return (a lottery ticket) gave some pleasure.

### Funding
The work was supported by UiT—The Arctic University of Norway. The publication charges for this article have been funded by the publication fund of UiT—The Arctic University of Norway. The funders had no role in study design, data collection and analysis, decision to publish, or preparation of the manuscript.

### Grant Disclosures
The following grant information was disclosed by the authors:
UiT—The Arctic University of Norway.

### Competing Interests
The authors declare that they have no competing interests.

### Author Contributions
- Mari V. Busch conceived and designed the experiments, performed the experiments, analyzed the data, authored or reviewed drafts of the paper, approved the final draft, sampled all data for study 2, analyzed them and wrote up her ms thesis based on them.
- Sandra Olaisen conceived and designed the experiments, performed the experiments, analyzed the data, approved the final draft, sampled most of the data in study 1, analyzed them and wrote up her bachelor thesis.
- Ina Jeanette Bruksås conceived and designed the experiments, performed the experiments, analyzed the data, approved the final draft, sampled some of the data for study 1, analyzed them for her ms thesis.
- Ivar Folstad conceived and designed the experiments, analyzed the data, prepared figures, authored or reviewed drafts of the paper, approved the final draft, initiated the studies, supervised the three students and wrote up the ms.

### Human Ethics
The following information was supplied relating to ethical approvals (i.e., approving body and any reference numbers):

All data in the present study were anonymized and, according to the Norwegian Data Protection Office for Research (NSD), no ethical approval was needed (see http://www.nsd.uib.no/personvernombud/en/notify/notification_test.html)

## Data Availability

The data are available at UiT Open Research Data https://dataverse.no/dataverse/uit https://dataverse.no/dataset.xhtml?persistentId=doi:10.18710/DQJS4A

## Supplemental Information

Supplemental information for this article can be found online at http://dx.doi.org/10.7717/peerj.5924#supplemental-information.

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
