# Peer review of "Do mothers also “manipulate” grandparental care?"

_PeerJ, doi:10.7717/peerj.5924_

## Round 0.1 · original submission · Minor Revisions

Both referees liked the manuscript and I agree with them that this is a very nice piece of work. They had some comments to further improve the clarity of the manuscript, and spotted a few errors/typos that should be corrected in the final version.

Reviewer 1 ·

Basic reporting

See below. No further comments.

Experimental design

See below. No further comments.

Validity of the findings

See below. No further comments.

Additional comments

I reviewed this manuscript previously for another journal and I am happy to see that this is an improved version of the already satisfactory original version that I saw before. In this study the authors put forward a controversial and most interesting hypothesis, the “manipulative mother hypothesis”, according to which mothers (compared to putative fathers) exploit the alloparenting environment (namely grandparental investment) by expressing biased resemblance descriptions. The hypothesis makes sense from an adaptive point of view and in turn fits with the well-accepted notion that paternity uncertainty modulates grandparental investments: Maternal grandmothers, who have higher certainties of genetic relatedness with grandchildren are predicted to invest most in grandparental care, compared to other grandparents, and especially compared to paternal grandfathers, whose grandpaternal genetic links are the least certain of the four grandparents. To test their hypothesis the authors gather data from two questionnaires filled by two populations of Norwegian respondents. The findings are most revealing: daughters more often and more intensely than sons voice their opinions about the resemblance between their parents and their grandchildren, and the results in the questionnaires point towards the idea that this conscious or unconscious manipulation may bias grandparental investment.

The paper is well written, the study has good (internal) controls, and the authors make a strong case to consider seriously the controversial hypothesis that is assessed. The hypothesis complements and expands the paternity uncertainty hypothesis, and by doing so this manuscript makes a meaningful contribution to the field that no doubt will attract attention by the scientific community. Whether the “manipulative mother hypothesis” remains valid after further tests aiming to falsify its predictions remains to be seen, but both the concept and these first results are worth publishing.

I only have a few comments and several minor suggestions:

1. The authors justify why they do not adjust for multiple probability estimates. While what they say is true I do not think the justification of big data applies specifically to this study. However, I agree that the focus should be in effect size rather than p values. Perhaps the authors would like to reformulate their justification.

2. Reported frequency of contact can be a proxy for investment, but frequency of contact also depends on the grandparents’ sons or daughters’ manipulation abilities and many other factors (e.g., social status, working status, and so on), and biases in frequency of contact also depend on whether there are interactions between these factors and the sex of the grandparent and that of his/her offspring (e.g., grandmothers could be more susceptible to being manipulated than grandfathers, and daughters could be better at manipulating grandmothers etc.). Discussing at length all these possibilities would perhaps enter the realms of going too far with speculation, but
acknowledging that these kind of modulations may further play a role could be useful. This is not a major issue and it should be left to the authors’ decision.

3. The study’s results are based on responses to questionnaires, and this may come with noise and uncertainty stemming from many factors. However, the authors make a strong case for the interpretations, ensuring that some of the potential confounding factors are kept at bay, and they further acknowledge some limitations and provide satisfactory arguments against them. Participants in Study 2 are all academics from a University. However, the authors acknowledge these limitations and show why they are not serious shortfalls. Moreover, this kind of studies necessarily comes with some of these limitations, as manipulative experiments on these questions in humans are not feasible or ethical.

4. The study has good controls overall. For instance, the same individual is at the same time a maternal and paternal grandparent. Also, each respondent grandparent is independent from any other grandparent. For future studies the authors could consider that having different grandparents for same offspring-grandoffspring (and controlling for non-independence) would be interesting as well from a different perspective, and would be a conservative control. For instance, both grandparents in a couple might contribute equally because of manipulative actions by one of the two grandparents (e.g., by the maternal grandmother upon the maternal grandfather). Likewise, if data were available such that investment into the same two focal grand-offspring (the grandson and the granddaughter) by all the actors involved (i.e., all the 4 grandparents) could be assessed, that would be informative. This is out of the scope of the present study but they could be the basis of useful avenues to further refine some of the findings in the present study. As before, this is a just a comment to the authors and requires no action in the present manuscript.

5. Other comments:
- Some references in the list may require attention. For instance, reference 31 is in capital letters.
- Please provide a citation for the statement that the bias of mothers ascribing higher resemblance of newborns to the mothers’ partner (putative father) than to themselves is exaggerated in the presence of the putative fathers and reduced in their absence (line 120). References are given before for maternal manipulation but a specific referral to the research providing the exaggeration of that bias, which is very revealing, would be welcome in this line.
- L517. fMRI acronym is not explained unless I missed it.

·

Basic reporting

The manuscript is very well written and very well referenced with respect to the literature. The statistical analysis is straightforward and easy to understand. The figures included are useful.

Experimental design

The experimental design is both good and novel. The problem is clearly stated and the statistical analysis is adequate.

Validity of the findings

While the sample sizes are generally small, some robust significant results of general interest were obtained. The authors discussion of their results in sound, supported logically, and is not overly speculative.

Additional comments

Comments on MS #30207: Do mothers also manipulate grandparental care?
In this manuscript, the authors use questionnaires to determine access of grandparents to grandchildren and to determine if the parents of grandchildren differ by sex in how they ascribe the resemblance of their children to the children’s grandparents. The authors argue that their results are not consistent with paternity uncertainty and posit that mothers attempt to manipulate children’s grandmothers to their own reproductive advantage. The use of personality as well as physical resemblance is nice.
The manuscript is very well written and very well referenced with respect to the literature. The statistical analysis is straightforward and easy to understand. The figures included are useful. I did find the results a little difficult to keep straight as there were many analyses pertaining to the questionnaires and it was not always apparent that a statistical test supported a statement. I suggest a figure for each questionnaire that is a flow chart showing the questions and other information relevant to each relative. Parts of each flow chart could be numbered to correspond to statistical tests that are then placed in tables. Thus, the results section would contain only statements about what was found (not test results) and readers could match statistical tests to parts of the questionnaires. Subheadings in the discussion section would also help readers follow arguments.
I agree that the uncertainty of paternity hypothesis fails to account for the data. This hypothesis would seem to predict that ascriptions of resemblance should be made especially to paternal grandparents and even made by the fathers of those grandchildren. The manipulative mother hypothesis seems reasonable but also does not account for the indifference of the fathers of the grandchildren. Perhaps, part of the explanation lies in the social structure of ancient societies. If a small number of males had harems, paternal grandparents could have many grandchildren relative to maternal grandparents, reducing their ability to invest in individual grandchildren. Males may then have evolved not to bother with appeals to their own parents in their children’s behalf. Maternal grandparents would often have fewer grandchildren and might, therefore, be better able to assist them. I admit that I may not have thought this through. The authors may consider this point if they so desire.
I noticed a few typographical errors. These are: (1) Line 153, add “s” to “material” and to “method”, (2) Line 228, change “do” to “does”, (3) Line 286, add subheadings to match those of experiment 1, (4) line 292, add “is” after “This”, (5) Line 373, change “were” to “was”, (6) Line 374, the use of “children” seems wrong, (7) Line 429, add an “s” to “suggest”, and (8) Line 454, change “believes” to “beliefs”.

---

## Round 0.2 · accepted · Accept

All the minor points raised by the two referees during the previous round of review have been satisfactorily addressed. Thank you for this nice piece of work.

#